

# Estimation of atmospheric particle formation rates through an analytical formula: Validation and application in Hyytiälä and Puijo, Finland

Elham Baranizadeh[1], Tuomo Nieminen[1], Taina Yli-Juuti[1], Markku Kulmala[2], Tuukka Petäjä[2], Ari
Leskinen[1,3], Mika Komppula[3], Ari Laaksonen[1,4], Kari E. J. Lehtinen[1,3]

[1]Department of Applied Physics, University of Eastern Finland, Kuopio, Finland

[2]Department of Physics, University of Helsinki, Finland

[3]Finnish Meteorological Institute, Kuopio, Finland

[4]Climate research Unit, Finnish Meteorological Institute, Helsinki, Finland

*Correspondence to*: Elham Baranizadeh (elham.baranizadeh@uef.fi)

**Abstract.** The formation rates of 3-nm particles were estimated at SMEAR IV, Puijo (Finland) where the continuous measurements extend only down to 7 nm in diameter. We extrapolated the formation rates at 7 nm ($J_7$) down to 3 nm ($J_3$) based on an approximate solution to the aerosol general dynamic equation, assuming a constant condensational growth rate, a power-law size dependent scavenging rate and negligible self-coagulation rate for the nucleation mode particles. To evaluate our method, we first applied it to new-particle formation (NPF) events in Hyytiälä (Finland), which extend down to 3 nm, and, therefore, $J_3$ and $J_7$ can be determined directly from the measured size distribution evolution. The Hyytiälä results show that the estimated daily mean $J_3$ slightly overestimate the observed mean $J_3$, but a promising 84% of the estimated $J_3$ are within a factor of 2 from the measured ones. However, when considering detailed daily time evolution, the agreement is typically poor presumably due to uncertainties in estimated growth rates which are required in order to calculate the time-lag between formation of 3-nm and 7-nm particles. At Puijo, the mean and median $J_7$ for clear NPF days during April 2007-December 2015 were 0.23 and 0.07 cm$^{-3}$s$^{-1}$, respectively, while the extrapolated mean and median $J_3$ were 0.47 and 0.13 cm$^{-3}$s$^{-1}$, respectively.

## 1 Introduction

Atmospheric new particle formation (NPF) events, i.e. nucleation and subsequent growth of newly formed particles have received increasing attention due to their impact on climate and human health (Kulmala et al., 2004; Merikanto et al., 2009; Nie et al., 2014, Kerminen et al., 2012; Fuzzi et al., 2015, Minguillón et al., 2015 and references therein). Many studies have been conducted to find out which variables cause and which possibly inhibit NPF events. Sulfuric acid, water and ammonia



have already long been considered important molecules for atmospheric new particle formation (Weber et al., 1995; Weber et al., 1996; Korhonen et al., 1999; Kulmala et al., 2000; Laaksonen et al., 2008; Xiao et al., 2015). More recently, studies
show that amines, ions, and oxidation products of volatile organic compounds can play an important role in NPF events either by participating in the nucleation itself or by stabilizing the nucleated clusters (e.g. Almeida et al., 2013; Berndt et al., 2014; Kirkby et al., 2016). However, several features at the nucleation level including the actual mechanism and other possible vapors involved (Kulmala et al., 2006; Lehtinen et al., 2007) remain unknown.

The lack of exact knowledge of NPF mechanisms is partly because at several locations particle size distribution measurements do not extend to nucleation size range but instead start at ca. 3 nm or even at larger sizes (e.g. 7 or 10 nm). This limits the use of the particle data in NPF studies and poses a challenge in understanding NPF globally. In addition, the actual nucleation rates of critical clusters sizes (sub-2-nm in diameter) remain unknown. Even with data obtained by the new condensation particle counters (CPC), that have cut-off mobility diameters of sub-2 nm (Sgro and Fernández de la Mora,
2004; Iida et al., 2009; Vanhanen et al., 2011; Kuang et al., 2012; Wimmer et al., 2013), the determination of nucleation rates still involves approximation.

Measuring sub-3-nm particles is a challenging task because of their diffusion loss during transporting the sample, difficulties in collecting representative samples for electrical detection, difficulties in charging them for electrical size-selection
(classification), their insufficient amount to be chemically analyzed, and the need for a very high supersaturation condition to grow them to large enough sizes that they can be optically detected (Kulmala et al., 2012). Because of these challenges in measuring small particles, methods to extrapolate size distributions and formation rates below the measurement range have been suggested by McMurry and Friedlander (1979), McMurry (1982; 1983), Weber et al. (1996); Kerminen and Kulmala (2002); Kerminen et al. (2003); Lehtinen et al. (2007) and most recently by Kürten et al. (2015). We are, however, not aware
of another study in which these methods have been tested with atmospheric measurement data.

Our study has two main goals. Firstly, we aim to estimate 3 nm particle formation rates $J_3$ for Puijo, where continuous size distribution measurements have been going on since 2006. We estimate the $J_3$ by a scaling method based on aerosol dynamics theory for the range 3 - 7 nm, because the measured size range at Puijo has been only down to 7 nm in diameter.
Therefore, our second main goal is to validate our method to estimate $J_3$. For this, we use size distributions measured at Hyytiälä, where detailed particle size distribution measurements down to 3 nm have been performed since 1996. From the Hyytiälä data we can thus evaluate formation rates both at 3 nm and 7 nm. The fraction of particles that survives the scavenging by larger aerosols is determined by the ratio of their growth and scavenging rates (Kerminen et al., 2004b). In this study, we use the method of Lehtinen et al. (2007) in which time and size independent particle growth rate and, time
independent but size dependent coagulation sink are assumed.



## 2 Methods

### 2.1 Data sets and site descriptions

In this study we use the aerosol size distribution measurements at two different SMEAR (Station for Measuring Ecosystem-Atmosphere Relations) stations in Finland: SMEAR II located in Hyytiälä and SMEAR IV in Kuopio. SMEAR II (Hyytiälä, southern Finland; 61°51′ N, 24°17′ E, 181 m a.s.l.) is characterized by boreal coniferous forest. The main pollution sources are the city of Tampere (60 km away) and the buildings at the station. These sources are most effective when the wind is from the southwest direction (Kulmala et al., 2001). For this study we analyzed aerosol size distributions measured at SMEAR II with a Differential Mobility Particle Sizer (DMPS; Aalto et al., 2001), with a cut-off size at 3 nm, between years 2000-2012.

At SMEAR IV the instruments are set up at the top of the Puijo observation tower (62°54′34″ N, 27°39′19″ E), 306 m and 224 m above the sea level and the surrounding lake level, respectively). Puijo tower is located in the city of Kuopio (Eastern Finland), a semi-urban environment with surroundings characterized by forest with conifer and deciduous (mostly birch) trees, and many lakes. The main local sources surrounding the tower are a paper mill (direction 35°, distance >1.4 km), the city center (direction 120-155°, distance 1.6-3.2 km), a heating plant (direction 160°, distance 3.5 km), a highway and residential areas (see Leskinen et al. (2009) and Portin et al. (2014) for more details). The aerosol size distribution is measured with a twin-DMPS (Winklmayr et al., 1991; Jokinen and Mäkelä, 1997) covering the size range 7-800 nm (Leskinen et al., 2009). The twin-DMPS consists of two differential mobility analyzer (DMA) tubes, one shorter with 11-cm length and another one longer with 28-cm length, and a condensation particle counter (TSI Model 3010 CPC) after each DMA tube. In both DMPS systems, the sample is neutralized (before it enters to the DMA) into charge equilibrium by a beta radiation source (Ni-63 10 mCi=370 MBq). The size range measured by the longer tube is 27-800 nm with 29 discrete bins and 7-49 nm with 17 discrete bins for the shorter tube. The full particle size distribution (7-800 nm) is measured every 12 minutes (Leskinen et al., 2009). At Puijo there is a twin-inlet system for aerosol-cloud interaction studies: one inlet removes cloud droplets (when the station is in a cloud) and collects only the interstitial particles and the other inlet collects the total aerosol, i.e. cloud droplets and interstitial particles. When the station is not in a cloud, the size distribution measured from both inlets are the same. In this study, we used the data from the total aerosol inlet and analyzed aerosol size distributions measured between April 2007 and December 2015.

### 2.2 Data analysis method

Kerminen and Kulmala (2002) derived an analytical formula which links the "real" particle formation rate and the "apparent" formation rates of particles of larger sizes for which measurements are available (typically above 3 nm). The formula was later improved by Lehtinen et al. (2007) by (1) correcting the slightly inaccurate size dependence of the





coagulation sink, and (2) removing the unnecessary assumption of the identity of the condensing vapor. According to the formula (equation (7) in Lehtinen et al., 2007) one can estimate the formation rate of smaller particles ($J_{d1}$) with diameter $d_1$, for which no measurements are available, from formation rate of measured larger particles ($J_{d2}$) with diameter $d_2$, as

follows:

$$J_{d1} = J_{d2} \cdot \exp\left(\gamma \cdot d_1 \cdot \frac{CoagS(d_1)}{GR}\right),$$ (1)

with $\gamma = \frac{1}{m+1}\left(\left(\frac{d_2}{d_1}\right)^{m+1} - 1\right)$ and $m = \frac{\log[CoagS\,(d_2)/CoagS(d_1)]}{\log[d_2/d_1]}$,

where $CoagS$ is the coagulation sink of smaller particles (diameter $d_1$) onto the background particles, and $GR$ is the particle growth rate (which is assumed to be constant from diameter $d_1$ to diameter $d_2$).

In this study, we apply the Eq. (1) to estimate the apparent formation rates of particles of 3 nm in diameter at Puijo where the size distribution of particles below 7 nm is not measured. To derive Eq. (1) (i.e. equation (7) in Lehtinen et al., 2007), it was

assumed that the growth rate between $d_1$ and $d_2$ is constant. This assumption, however can fail especially for sizes below 3 nm, where some recent studies have indicated strong size dependence of $GR$ (Kuang et al., 2012; Kulmala et al., 2013).

Korhonen et al. (2014) modified Eq. (1) to also include either linear or power-law type size dependent growth rate and tested the method by using modelled NPF events. In their studies especially the method assuming power-law type growth rate gave

promising results with various types of size dependent growth profiles. However, in this study, we assume a constant $GR$ because as mentioned earlier a strong size-dependency of $GR$ has been reported for very small particles typically below 3 nm (e.g. Kuang et al., 2012) rather than for larger sizes. The other assumption when deriving Eq. (1) is that the nucleating particles are lost only by coagulation onto larger pre-existing particles.

To evaluate Eq. (1) against measurements, we use the particle size distribution evolution data during nucleation event days from SMEAR II. There the measurements have extended down to 3 nm in diameter, and therefore, one is able to get apparent formation rates at 7 nm ($J_7$) and at 3 nm ($J_3$) directly from measurements. We then set $d_1 = 3$ nm and $d_2 = 7$ nm in Eq. (1) and calculate $J_{3,obs}$ and $J_{7,obs}$ as outlined in Kulmala et al. (2012) and slightly improved in Vuollekoski et al. (2012). Here we use the subscript *obs* to indicate *observed* apparent formation rates $J$. The formation rate of particles of 3 nm ($J_{3,obs}$) and 7

nm ($J_{7,obs}$) in diameter from measured aerosol size distribution were calculated as follows:

$$J_{3,obs} = \frac{dN_{3-7}}{dt} + n_7 \cdot GR_{7-20}, + N_{3-7} \cdot CoagS(d_{GMD}),$$ (2)



where $n_7 = \frac{N_{5-9}}{9-5}$ and $d_{GMD} = \sqrt{3 \times 7}$ nm.


$$J_{7,obs} = \frac{dN_{7-10}}{dt} + n_{10}.GR_{7-20} + N_{7-10}.CoagS(d_{GMD}),\tag{3}$$

where $n_{10} = \frac{N_{8-12}}{12-8}$ and $d_{GMD} = \sqrt{7 \times 10}$ nm.

Here $N_{3-7}, N_{5-9}, N_{7-10}$ and $N_{8-12}$ are the number concentration of particles within size ranges 3-7 nm, 5-9 nm, 7-10 nm and 8-12 nm, respectively, and $n_7$ and $n_{10}$ are the size distribution function at 3 nm and 7 nm, respectively. The coagulation sink ($CoagS$) terms were calculated directly from the measured particle size distributions, taking into account the hygroscopicity effects using the parametrization of Laakso et al. (2004) who used the hygroscopic growth factor parametrization by Zhou (2001). We used a parabolic differentiation method to the measured number concentration to obtain its time-derivative (the

first term in Eq. (2) and Eq. (3)). The method fits a second order polynomial to seven data points centered at the data point where derivative is calculated while at the edges a parabola is fit through the first or last six data points, from which the derivative is calculated directly. Also, to avoid spurious fluctuations in the second and third terms in equations 2 and 3, the $N_{3-7}, N_{5-9}, N_{7-10}$ and $N_{8-12}$ were smoothed using a moving average (over five data points) filter.

The estimated formation rate $J_3$ was then calculated based on Eq. (1):

$$J_{3,est}(t) = J_{7,obs}(t') . \exp\left(\gamma(t).3nm.\frac{CoagS(d_1=3nm)}{GR_{3-10}}\right),\tag{4}$$

Note $J_{3,est}$ at time $t$ is calculated based on $J_{7,obs}$ at time $t'$, where $t = t' - \frac{4nm}{GR_{3-10}}$, thus accounting for the growth time of the

3 nm particles to 7 nm particles. To average over this time interval needed for growth, the $m$ and $CoagS(d_1)$ values are calculated as medians of the corresponding values during time $t$ to $t'$.

To determine the growth rates required in this study, we first used the automated algorithm developed by Hussein et al. (2005) for fitting log-normal modes to the measured size distributions. The algorithm assumes that the size distribution is a

superposition of 1-3 log-normal modes and at each measurement time optimizes three unknown parameters for each mode to fit the measurements. The parameters for each individual log-normal mode are the mode number concentration $N_i$, geometric variance $\sigma_g^2$, and geometric mean diameter $D_{pg}$. We then estimate $GR$ by fitting the geometric mean diameter $D_{pg}$ of the growing nucleation mode as a function of time; the slope of the fitted line determines the $GR$ in the desired particle size ranges. We also determined the standard error ($SE$) of the $GR$ estimates when fitting $D_{pg}$ values respect to time to obtain $GR$.

We left out the days where the growth rates required in the aforementioned equations (i.e. $GR_{3-10}$ and/or $GR_{7-20}$) were not





quantifiable. We chose the size range 3-10 nm rather than 3-7 nm to determine the $GR$ in the exponential term of equation 4 (denoted as $GR_{3-10}$). This was done to increase the number of data points in the $GR$ fitting and thereby to improve the reliability of the fitted $GR$.

After evaluating the analysis method with SMEAR II data, we applied the method for Puijo where the DMPS detection range extended only down to 7 nm. To estimate the formation rate of 3-nm particles at Puijo we adapted Eq. (4) by replacing $GR_{3-10}$ with $GR_{7-20}$ due to lack of DMPS measurements below 7 nm. However, as it will be shown in section 3.1, using $GR_{7-20}$ instead of $GR_{3-10}$ does not affect the accuracy of estimated $J_3$ for NPF events in Hyytiälä, which is an indication that the size dependence of the growth rate in the range 3-20 nm is typically weak. The $J_{7,obs}$ was calculated with the same

method as was used for Hyytiälä (i.e. using equation 3).

## 3 Results and discussion

### 3.1 Analysis of estimated $J_3$ in Hyytiälä (Finland)

Figure 1 shows the comparison of estimated formation rates $J_{3,est}$ (Eq. (4)) with the observed ones $J_{3,obs}$, as calculated directly from the measured size distribution evolution according to Eq. (2) in Hyytiälä. We analyzed 65 NPF event days for

which the formation and growth rates could be quantified. Each data point in Figure 1-a represents the arithmetic mean of the 3-nm particle formation rates ($J_{3,est}$ and $J_{3,obs}$) for a single NPF day during the time window from 07:00 to 19:00 local time. The results show that the estimated mean $J_{3,est}$ values agree reasonably well with $J_{3,obs}$ with correlation coefficient 0.78 and 85 % of estimated $J_{3,est}$ are within the factor of two of the observed $J_{3,obs}$. Equation (4) seems to have a tendency of overestimating the formation rate of 3-nm particles. We calculated the arithmetic mean of all data points (the total mean)

presented in Figure 1-a. The total means of $J_{3,obs}$, $J_{3,est}$ and $J_{7,obs}$ (not shown in the figure) are 0.21, 0.27 and 0.14 # cm$^{-3}$ s$^{-1}$, respectively, confirming the tendency of Eq. (4) in overestimating the 3-nm particle formation rates. The color code of Figure 1-a indicates the ratio of the relative standard error (SE) of $GR_{3-10}$ and $GR_{3-10}$ (i.e. $SE/GR_{3-10}$). According to Figure 1-a no relationship between uncertainty in $GR$ estimates and formation rate estimates is seen. For example the data points close to the 1:1 line consist of both days with high and days with low values for $SE/GR_{3-10}$.


Moreover, we also compared the estimated and observed $J_3$ values using daytime median values (not shown here), resulting in a correlation coefficient of 0.73 between $J_{3,obs}$ and $J_{3,est}$. The median $J_{3,est}$ values are, however, even more overestimated than the corresponding mean values and only 38% of estimated $J_{3,est}$ are within a factor of two of the observed $J_{3,obs}$. Total medians (median of daily-median values) of $J_{3,obs}$, $J_{3,est}$ and $J_{7,obs}$ are 0.05, 0.106 and 0.047 # cm$^{-3}$ s$^{-1}$.




In addition, we replaced $GR_{3-10}$ with $GR_{7-20}$ in Eq. (4) as will be needed to estimate $J_{3,est}$ in Puijo. Results show that although the correlation coefficient improves to 0.93, a smaller fraction (78%) of $J_{3,est}$ data points are within the factor of two of $J_{3,obs}$ values, and subject to more bias (overestimation). However, these changes are minor and do not significantly affect the results. The total mean of $J_{3,est}$ changed from 0.27 to 0.31 # cm$^{-3}$ s$^{-1}$ after replacing $GR_{3-10}$ with $GR_{7-20}$ in Eq. (4).


Furthermore, we also tested how replacing $GR_{3-10}$ with $GR_{3-7}$ in Eq. (4) affected the estimated $J_{3,est}$ values. Replacing $GR_{3-10}$ with $GR_{3-7}$ resulted in similar bias (i.e. towards overestimation) and agreement between $J_{3,est}$ and $J_{3,obs}$ mean values, to what was obtained from replacing $GR_{3-10}$ with $GR_{7-20}$: the correlation coefficient slightly improved (0.82) with slightly less $J_{3,est}$ data points (80%) within the factor of two of $J_{3,obs}$. In general replacing $GR_{3-10}$ with $GR_{3-7}$ did not affect the results (i.e.

agreement level between of $J_{3,est}$ and $J_{3,obs}$) by much.

Figure 1-b shows $J_{3,obs}$ versus $J_{3,est}$ values with the same 10-minute temporal resolution as for the measured size distribution. The points are within the time window from 07:00 to 19:00 local time. With this higher temporal resolution $J_{3,obs}$ and $J_{3,est}$ are not correlated (correlation coefficient = 0.17) despite that their daily mean values presented in Figure 1-a

correlate clearly; only 32% of the estimated $J_{3,est}$ are within factor of two of the observed $J_{3,obs}$. The main reason for this is the lack of success in estimating the time lag between the formation of 3-nm and 7-nm particles (see for example the Figure 3-b presented later in this section), which then results in an incorrect time shift for the time evolution of $J_3$, even though the daily average values agree reasonably well.

After replacing $GR_{3-10}$ with $GR_{7-20}$ in Eq. (4), still 31% of estimated 10-minute $J_{3,est}$ are within factor of two of the observed $J_{3,obs}$, correlation coefficient slightly worsens to 0.13 and, the $J_{3,est}$ data are subject to more bias (positive bias thus overestimation). We, therefore, conclude that replacing $GR_{3-10}$ with $GR_{7-20}$ both for mean $J_{3,est}$ values and 10-minute values, has only a minor effect on the results thus using $GR_{7-20}$ to estimate $J_3$ values in Puijo is reasonable.

Figure 2 shows examples of the time evolution of the particle size distribution as well as the different formation rates $J$ on three NPF days in Hyytiälä. For some NPF days, the estimated time-dependence (or time-lag between 3-nm and 7-nm particle formation rates) and values of $J_{3,est}$ are in fairly-good agreement with those of observed $J_{3,obs}$ (see e.g. Figure 2-d). However, the time-dependency of $J_{3,est}$ is not consistent with $J_{3,obs}$ for most of the days and, instead, typically the $J_{3,est}$ peak occurs earlier than the $J_{3,obs}$ peak (see e.g. Figure 2-e), indicating that our method of estimating $GR$ is not satisfactory and

typically understimates the $GR$ values. In order to investigate how well Eq. (4) estimates the time evolution of the 3-nm particles we visually chose the days during which a clear peak in each of the $J_{7,obs}$, $J_{3,obs}$ and $J_{3,est}$ time evolution curves could be observed (39 days out of 65 days). For these events, we extracted the time difference between 3-nm and 7-nm particle formation from 1) the observed time between peaks in $J_{3,obs}$ and $J_{7,obs}$ (named here observed time-lag), and 2) from





growth time $t' - t = 4 \text{ nm}/GR_{3-10}$ (named estimated time-lag) which is also equal to the time difference between $J_{3,est}$ and

$J_{7,obs}$. Figure 2-f shows an example of a NPF day for which the $J_{3,est}$ and $J_{3,obs}$ are dramatically different. This is due to the

burst in the number concentration which appeared mostly within the size range 3-7 nm (chosen to calculate $J_{3,obs}$) and is thus

not included in the size range 7-10 nm from which $J_{7,obs}$ is calculated and then scaled to $J_{3,est}$. Therefore, Eq. 4 can give

quite inaccurate results for NPF days associated with e.g. this type of inhomogeneity in the particle number concentrations in

different size ranges. It can be also concluded that visual inspection of the data is still valuable - cases like this are very

challenging for automatic data analysis routines.  Figure 3 shows the estimated time-lag versus the observed time-lag. As can

be seen from the figure, the estimated time-lag is mostly longer (toward earlier times of $J_{3,obs}$) than the observed time-lag.

There are 15 NPF days for which the estimated time-lag is within 1.5 hours of the observed time-lag. Overall these results

from analyzing Hyytiälä data show that Eq. (4) can be used to estimate the mean formation rates of 3-nm particles with

reasonably good accuracy. However, the performance in predicting detailed time-evolution of the 3-nm particle formation

rate is poor in most NPF days with the methods that we use for $GR$ estimation.

### 3.2 Estimation of $J_3$ in Puijo (Finland)

For the aerosol size distribution data in Puijo, the NPF event days were first recognized visually and classified as

"quantifiable" and "non-quantifiable" based on whether or not the event is homogeneous enough to allow quantification of

the basic characteristics such as formation and growth rates (Dal Maso et al., 2005). Therefore, our data pool consists of

event (E), non-event (NE) and undefined days, the last being days during which the evolution of the size distribution is too

unclear for definitive determination of whether or not NPF has been occurring. We noticed that there are two types of

undefined days in Puijo. One is characterized with a burst in the number concentration of particles of the smallest detectable

sizes but doesn't seem to show the characteristics of a NPF event day (i.e. growth to larger sizes, see e.g. Figure 4-a) and

most likely originate from local emissions. In the other type, some particles appear in larger sizes (with minor growth),

which may or may not be originated from NPF processes. (e.g Figure 4-b) like the first type. Note that 48 and 44% of the

days are missing during years 2010 and 2012, respectively. The monthly number and yearly fraction of NPF event days

recorded in Puijo from year 2007 to 2015 are shown in Figure 5. The figure shows that a maximum number of event days

occurred during spring time similar to NPF events reported in Hyytiälä (Dal Maso et al., 2005). It is also worth noting that

the fraction of event days is monotonically increasing from 2012 to 2015. There are 75 quantifiable NPF event days for

which we calculated the $J_{3,est}$ at Puijo. Figure 6 shows the seasonal medians of $J_{3,est}$ and $J_{7,obs}$, $GR_{7-20}$ and coagulation sink

for 7-nm particles ($CoagS(d=7 \text{ nm})$) for the quantifiable NPF event days in Puijo. The total mean and median of $J_{3,est}$ are

0.47 and 0.13, respectively, while the corresponding values for $J_{7,obs}$ are 0.23 and 0.07 #cm$^{-3}$s$^{-1}$, respectively. Total means of

$GR_{7-20}$ and $CoagS$ of 7-nm particles for NPF days are 2.34 nm/h and 1.5×10$^{-4}$ 1/s, respectively. Thus, the mean $GR$ at Puijo

is somewhat lower compared to Hyytilä where mean value of $GR$ = 4.3 nm/h is reported for period April 2003- December

2009 (Yli-Juuti et al., 2011).



Table 1 summarizes the seasonal means of parameters presented in Figure 6. The seasonal mean 3-nm particle formation rates seem to have the highest values during spring (0.52 #cm$^{-3}$ s$^{-1}$ for 50 NPF days) and summer (0.53 #cm$^{-3}$ s$^{-1}$ for 12 NPF days) and drops significantly in fall. The seasonal median of the growth rate has its maximum in summer (4.14 nm/h) and minimum in spring (2.30 nm/h). The seasonal median *CoagS* values, however, seem to be rather constant in Puijo in contrast to Hyytiälä.

## 4 Conclusions

In this study, the formation rates of 3-nm particles in SMEAR IV, Puijo (Finland) were estimated. The measurements at Puijo extend only down to 7 nm in diameter, which means that we had to extrapolate to 3 nm using aerosol dynamics theory. The approach used here is based on the competing processes of condensational growth and scavenging onto background aerosols, assuming time and size independent growth rate and time independent coagulation sink in the range 3 to 7 nm.

To first evaluate our extrapolation method, we applied it to particle formation events at Hyytiälä, where DMPS measurements extend down to 3 nm and formation rates at 3 nm ($J_{3,obs}$) and 7 nm ($J_{7,obs}$) can thus be determined directly from the measured size distribution evolution. The results show that the estimated daily mean values of $J_3$ are in reasonably good agreement with observed mean $J_3$, with 84% of the estimated $J_3$ within a factor of two from the measured ones and, mostly overestimated. However, when considering detailed daily time evolution, the agreement is typically poor. This is caused by the fact that there is a time lag between $J_3$ and $J_7$ and to take this into account in the comparison an estimation of the growth rate $GR$ is needed. Estimating $GR_{3-10}$, as was shown from Hyytiälä data, does not seem to give satisfactory results for this purpose. It should be noted that we have to estimate $GR$ from the data above 7 nm for Puijo site due to the lack of the measured data below 7 nm.

At Puijo, the mean and median of $J_7$ for quantifiable particle formation days were 0.23 and 0.07 cm$^{-3}$s$^{-1}$, respectively, while the extrapolated mean and median $J_3$ were 0.47 and 0.13 cm$^{-3}$s$^{-1}$, respectively. These are about two times greater than the corresponding values in Hyytiälä. Asmi et al. (2011) reported monthly mean 7-nm particle formation rate between 0.1 and 0.2 #cm$^{-3}$s$^{-1}$ for the NPF events in the sub-Arctic Pallas station, Finland. The ultimate aim of this work is to predict nucleation rates from size distribution measurements that do not extend to sizes lower than 7nm. The results obtained in this study suggest this is very challenging, in large part due to the difficulty in reliably predicting the growth rate down to around 1.5nm. It is noted that the possible size dependence of this growth rate further complicates the matter.

**Acknowledgments:**

We gratefully acknowledge the financial support by the Academy of Finland Center of Excellence program (project numbers 272041, 1118615), the Nordic Centre of Excellence CRAICC, University of Eastern Finland (UEF) strategic funding, and ACTRIS (under the European Union Seventh Framework Programme (FP7/2007–2013) grant agreement no. 262254 and Horizon 2020 research and innovation programme under grant agreement no. 654109).

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





**Table 1: Seasonal means of observed formation rate of 7-nm particles ($J_{7,obs}$), estimated formation rate of 3-nm particles ($J_{3,est}$), growth rate of particles in nucleation mode size range 7-20 nm ($GR_{7-20}$) and coagulation sink of nucleation mode particles onto larger particles (CoagS(d=7 nm)) for NPF days which occurred at Puijo during Apr 2007 - Dec 2015. Note that because there is only one NPF event which occurred in winter, we excluded winter from the table.**

|  | $J_{7,obs}$ (#cm$^{-3}$ s$^{-1}$) | $J_{3,est}$ (#cm$^{-3}$ s$^{-1}$) | $GR_{7-20}$ (nm/h) | CoagS(d=7 nm) (1/s) |
|---|---|---|---|---|
| Spring (Mar-May) | 0,53 | 0,24 | 2,30 | $1.8 \times 10^{-4}$ |
| Summer (Jun-Aug) | 0,52 | 0,30 | 4,14 | $2.3 \times 10^{-4}$ |
| Fall (Sep-Nov) | 0,22 | 0,12 | 3,01 | $1.5 \times 10^{-4}$ |
| Overall | 0.23 | 0.47 | 2.70 | $1.9 \times 10^{-4}$ |



**Figure and figure captions**

**Figure 1: Estimated ($J_{3,est}$) against observed ($J_{3,obs}$) formation rates of 3 nm particles (#cm⁻³ s⁻¹) during new-particle formation (NPF) event days in Hyytiälä. Data points indicate a) 10-minute b) arithmetic mean $J_3$ between 07:00 to 19:00 local time for each NPF day. The color code indicates the ratio of standard error (SE) of $GR_{3-10}$ estimates through linear fitting and $GR_{3-10}$ itself .**



**The aerosol size distribution temporal resolution measured by DMPS at Hyytiälä is 10-minute. Note that the time-lag during which 3 nm particles grow to 7 nm particles is taken into account in the $J_{3,est}$.**

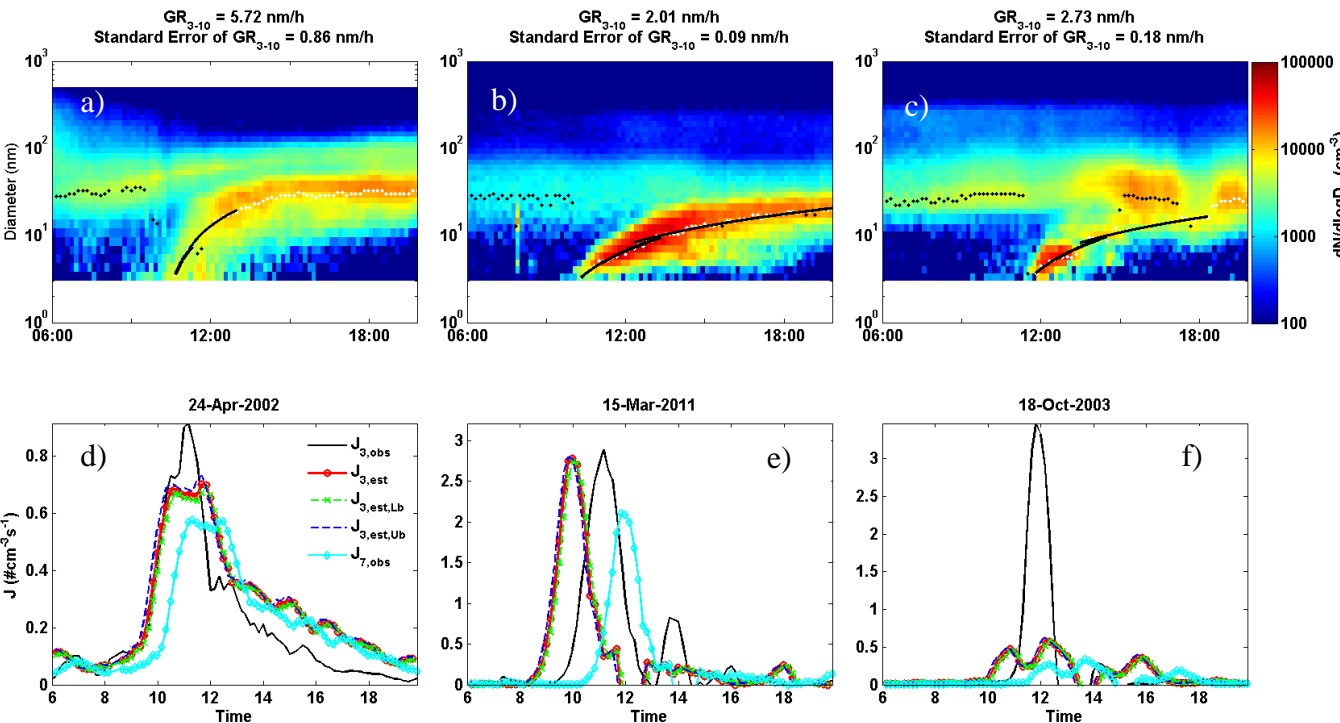

**Figure 2: Examples of Hyytiälä NPF events. a, b, and c) the evolution of the particle size distribution. White dots represent the geometric mean diameter of the nucleation mode determined by log-normal fitting, and the solid black line shows the first-order polynomial fit. Figures d), e) and f) are the corresponding evolution of 3 nm particle formation rates obtained from Eq. (2) (red), observed $J_{3,obs}$ (black) and observed formation rates of 7 nm particles $J_{7,obs}$ (cyan). The dashed curves show the upper bound ($J_{3,est,Ub}$) and lower bound ($J_{3,est,Lb}$) calculated using Eq. (2) 465 inputting the lower ($GR_{3-10} - SE$) and upper ($GR_{3-10} + SE$) bound of $GR_{3-10}$, respectively.**



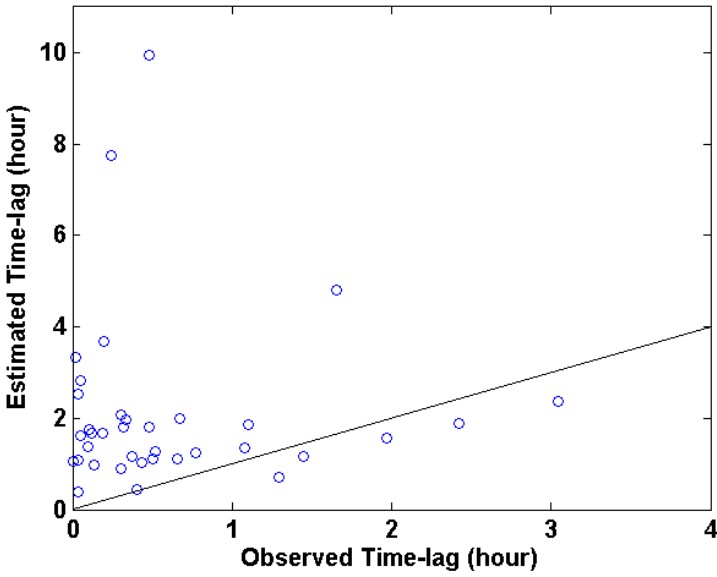

**Figure 3: Time lag between formation of 3 nm and 7 nm particles for NPF days at Hyytiälä determined visually from the time difference between the $J_{7,obs}$ and $J_{3,obs}$ peak (x-axis) and $t' - t = 4\ \text{nm}/GR_{3-10}$ (y-axis) which is also equal to the time difference between $J_{7,obs}$ and $J_{3,est}$.**

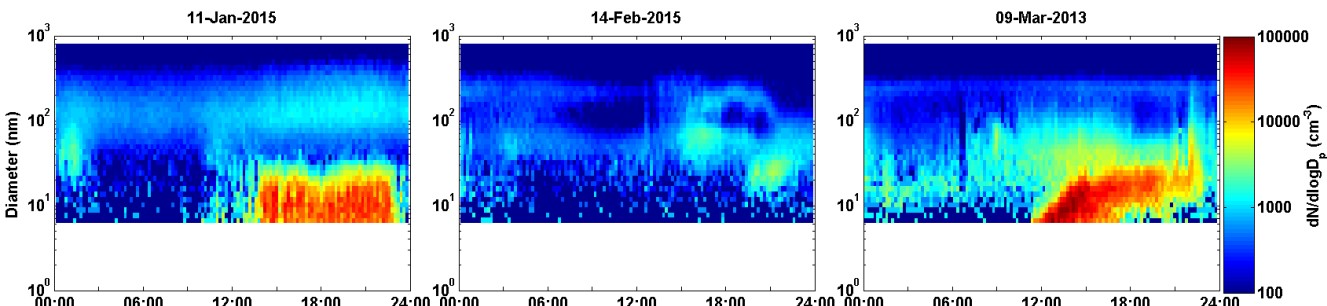

**Figure 4: Examples of the time evolution of the aerosol size distribution in Puijo for (a) an undefined day characterized by a burst in the number concentrations of the small particles which doesn't have the characteristics of a typical NPF event day (b) a typical undefined day, and (c) a clear NPF event day.**



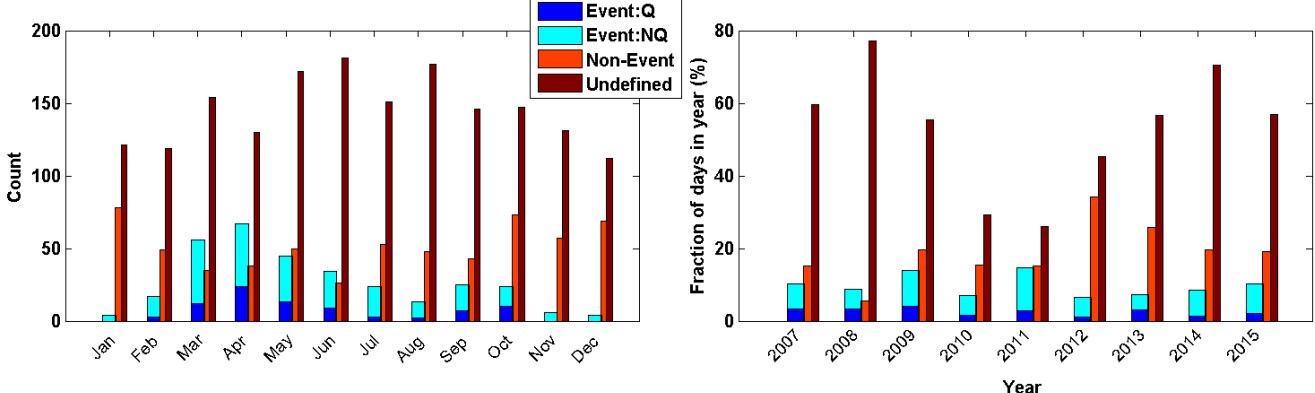

**Figure 5: Monthly number (left panel) and yearly fraction (right panel) of NPF event days (divided into Quantifiable Events (Q)**
**and Non-Quantifiable events (NQ)), Non-Events (NE) and undefined days recorded in Puijo during period 2007-2015. Fraction of**
**(e.g. NE) days in year is the ratio of number of NEs and number of days within the year. Note that the days for which bad or no**
**data were recorded are not shown here. Note that 48 and 44% of the days are missing during years 2010 and 2012, respectively.**

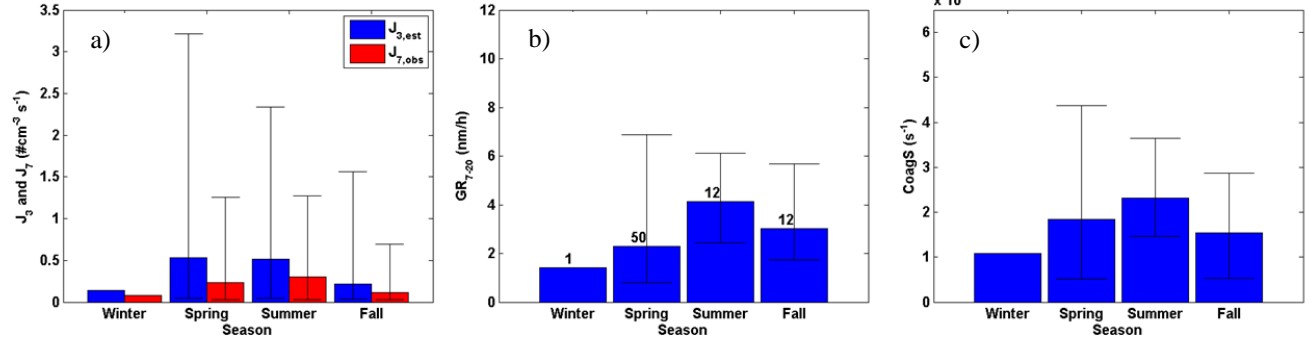

**Figure 6. Seasonal mean values of different parameters for NPF days at Puijo: a) estimated formation rates of 3-nm particles (**
$J_{3,est}$**) and observed formation rates of 7-nm particles ($J_{7,obs}$) b) growth rate of the particles within size range 7-20 nm c)**
**coagulation sink (*CoagS*) of 7-nm particles. The height of the bars shows the mean values of data points (i.e. mean values during**
**7:00 to 19:00 of the *J* and *CoagS* values for 75 NPF event days) within each season, and the error bars indicate the values between**
**minimum and maximum of the data points. The numbers on top of each bar in middle panel indicate the number of the NPF**
**events in corresponding season. The same applies to the figure 6-a and 6-c.**