# Peer review of "Estimation of atmospheric particle formation rates through an analytical formula: Validation and application in Hyytiälä and Puijo, Finland"

_Atmospheric Chemistry and Physics, 2016_

## Referee Comment (RC1) · Anonymous Referee #2 · 26 Jan 2017

GENERAL

The manuscript is basically suitable for ACP but a few important points need to be addressed first. I'll rate this "major revision" for now but remain skeptical that all concerns can be addressed to my satisfaction.

1. The authors present a method to estimate formation rates for smaller particles based on measurements of the formation rate for larger particles. This is no doubt necessary to compare measurements with different instruments and to gain information on the actual nucleation rate which happens at sizes which are often outside the measurement

range. However, a few years back, Kulmala et al. have published a sort of how-to guide on nucleation measurements in Nature Protocols. And that looks very, VERY similar to what is presented in this manuscript while the text reads as if something new is shown. So my question would be: What is actually new in the approach that the authors present? Or is this just another application of the same formula that has been used in lots and lots of papers for quite a number of years? If this were the case, the manuscript's content would be very slim indeed (and all the description of methods used obsolete) and one would have to ask if just running an old formula on a new set of data would justify a scientific publication.

2. The method to determine GR obviously doesn't work as the authors themselves point out. That is not a surprise since mode-fitting at the edge of a size distribution is always a bad idea. However, there are other methods. How can you justify using a method that clearly produces bad results while there are alternatives available? Sure, other methods typically are much more labour-intensive but given the current state of affairs it seems quite clear to me that other approaches MUST be employed. At least a preliminary test on a smaller subset of the data is absolutely and totally necessary.

3. I do wonder how there can be only 65 days with good enough data from 12 or 13 years of Hyytiälä observations. Assuming 12 full years à 365 days and an NPF frequency of 23% (Nieminen et al., 2014), that's a tiny 6.5% of all nucleation events observed during that time (about 1000). How can that be? Are we supposed to believe that one of the longest and probably the most published data set of aerosol size distributions is actually total crap? I mean, I have worked with DMPS and SMPS data quite a bit but never ever has the data been so terrible that a proper analysis was possible for less than 10% of events. And even if I was to accept this low percentage (which I can't and won't) then the question arises if this kind of cherry-picking doesn't introduce a bias into the analysis that would make all and any results highly questionable.

CONTENTS

line 31f "(e.g. Almeida et al., 2013; Berndt et al., 2014; Kirkby et al., 2016)" –> Bianchi et al., 2016 should probably be added there.

line 51 "we aim to estimate 3 nm particle formation rates" –> why? the 3 nm limit has no physical meaning, it's just a tradition born out of instrumental limitations from two decades back. i understand that you do that for the hyytiälä data since the point is testing the approach. but for puijo?

line 88ff –> this whole section is useless if this is the same approach as in the Nature Protocol

line 164 "the size dependence of the growth rate in the range 3-20 nm is typically weak" –> really? or is this just an artefact of the GR approach not working (which we know is true). certainly you could cite some previous studies that have found this; hyytiälä isn't exactly under-studied after all.

line 173 "85 %" –> wasn't it 84 in the abstract?

line 181ff –> the whole median thing seems a bit silly. i mean, you take the median over 12 hours during most of which there is no formation of 3 nm particles. Of course the result will be close to 0 (as it is). a pointless exercise which tells us nothing.

line 201ff, line 210ff, and lots of other places –> i won't comment on the GR stuff here, see general comments above.

line 216f "the days during which a clear peak in each of the [different] time evolution curves could be observed (39 days out of 65 days)" –> 39 out of 65 sounds quite good. but really it is 39 out of 1000, and that is not acceptable.

line 224 "It can be also concluded that visual inspection of the data is still valuable" –> that's sound advise that you might want to follow with regard to the GR business.

line 227 "There are 15 NPF days for which the estimated time-lag is within 1.5 hours of the observed time-lag." –> please take a moment and think what you have written

there. with an average GR of roughly 4 nm/h, the average time-lag should be around 1 hour, right? that 15 cases lie within 1.5 hours is no proof that the method works sometimes but rather that the method does not work AT ALL.

line 227ff "Overall these results from analyzing Hyytiälä data show that Eq. (4) can be used to estimate the mean formation rates of 3-nm particles with reasonably good accuracy." –> but maybe things could be much better with an improved determination of GR?

TECHNICAL

not necessary at this point

---

## Referee Comment (RC2) · Anonymous Referee #1 · 27 Jan 2017

General Comments:

A thorough, data-based evaluation of whether particle formation rates can be extrapolated from measurements at larger sizes, as attempted by this paper, is vital for the aerosol community as so many data exist with only larger size information available. While the method used to tackle this problem is valid and useful, the evaluation requires development and more nuanced analysis before the substantial conclusions stated in the paper can fairly be reached (see below for specific comments).

Specific Comments:

[Figure]

A major assumption, that the two measurement sites are directly comparable with the method used for extrapolating nucleation rates, is made in the paper. Kurten et al showed that the method used, while valid in many circumstances, may not be valid for situations where pre-existing populations of aerosols do not dominate the coagulation sink and newly formed particles play a larger role in this sink. The differences in background aerosols at the two sites should be discussed in relation to this. Differences between the two site may also influence the magnitude of growth rates and coagulation sinks, which may affect the accuracy of the J extrapolation, which should be addressed.

Line 59: The assumptions that the coagulation sink is time independent and the growth rate size independent should be more fully investigates. Kurten et al. highlights the possibility and affect of time dependent coagulation sinks. If this is not a problem for these two sites it should be explained more explicitly.

Line 75: some discussion of how the different environments of Hyytiala and Kuopio affect the average size distributions and patterns of nucleation would be helpful here. This can affect how accurately equation 1 can be applied. Equation 1 assumes that the coagulation sink is dominated by larger pre-existing populations, which is less applicable in cleaner environments. If the Hyytiala environment is much cleaner, for example, than Kuopio, then the two situations are less comparable for this method of calculating formation rates. This assumption is mentioned on line 114, but it's validity for both situations requires further discussion.

Line 145: averaging of m and CoagS(d1) between t and t' may be inaccurate, especially for high Js and low GRs – is the any indication of this in the data? How much do m and CoagS(d1) differ between t and t'?

Line 172: This discussion of how well J3,est and J3,obs agree needs further development. Suggest removing qualitative judgement of 'reasonably well', and leaving only quantitative measurements of this. While the 0.78 correlation coefficient is helpful, the

(linear?) fit result that this relates to would give a better measurement of the systematic difference between the two, this needs to be given here and on figure 1. This would then also quantify the following assertion that equation 4 overestimates the formation rate.

Line 175: Standard deviation should be given along with the daily means. These means are taken over a long period of time, during which I suspect J varies quite a bit. If J does vary a lot of this time period, then taking a daily mean is not very meaningful.

Line 184: The quoted daily median values of J are very small. Where in the day do they actually occur? Is it actually before a significant nucleation event occurs? If so these values are not very meaningful – suggest either cutting data to only encompass the nucleation event or finding a more meaningful statistic here.

Line 188: I would argue that reduction of percentage of points within a factor 2 of Jobs from 85% to 78% still reasonably significant and could indeed indicate that GR3-10 different and more accurate than GR7-20, which has strong implications for the conclusion that it's ok to use this GR in extrapolating results from Puijo. It would be more meaningful here to look at the fit equation again rather than simply correlation and percentage within factor 2 to understand this difference better.

Lines 191-195: Would prefer to see a full comparison of difference between observed and calculated Js here using GR3-10, GR3-7 and GR7-20, as well as a developed discussion of the degree of agreement and implications of this for using this method for J extrapolation. "Did not affect the results . . .by much" is too qualitative and glosses over what could be an important result here.

Lines196-203: The lack of correlation on temporarily resolved data here may indicate that the growth rates are wrong – this should be discussed here. It could also be because, by taking averages over long events where J varies significantly, the correlation seen early was simply an artifact of such heavy 'smoothing'. Is there another, meaningful measure of J (e.g. peak J of an event) that could be compared to asses

this?

Lines 205-208: Given the lack of correlation for time resolved Js, testing the affect of different GRs here does not have much meaning – suggest leaving this out completely.

Lines 211-212: "For some NPF days, the estimated time dependence and values of Jest are in fairly good agreement with those of observed Jobs." This statement needs better quantification to be of value. What proportion of days (since we're looking at a relatively small number of event, suggest quoting both total number of events examined and number of those with time dependence and value agreement here instead of just a percentage). How 'fairly good agreement' was judged needs explanation Also, are there distinguishing features of this sub-group where agreement is good? E.g. slow growth, classic 'banana' nucleation pattern?

Line 213: quantify 'most of those days'

Line 221: why does this burst of particles of 3-7nm occur and not then grow? Is this indicated by the calculated GR and coagulation sink? Or is it perhaps a transport artifact? If it is the later it should be removed from the analysis as it is not nucleation. If it's the former then the equation used to calculate J3 should be able to handle it. Therefore this needs full investigation and explanation.

Lines 225-226: Estimated time-lag longer than observed time-lag indicates that the GR used is too low, which has implications for the calculated J and the time-dependence of the nucleation event. Can this explain the poor ability of this method to reproduce the time-evolution of nucleation events? This should be discussed.

Line 227: 15 days out of how many in total? Line 227: 1.5 hours difference: what percentage of the total time lag is this?

Line 229: quantify 'reasonably good accuracy'

Line 244: This monotonic increase in number of event days per year with time is indeed worth noting. Is this because of improvements in instrumentation/data quality? Change

of activity or climate in the local area? Some discussion warranted. Do other things, such as total number of nucleation mode particles, size of coagulation sink, or anything else also monotonically change over this time period that might indicate why this is happening?

Line 245: Given the lack of correlation shown early between median J3 est and obs, using J3 est here for analysis does not seem justified. Surely mean J, where some correlation between estimated and observed values was calculated is the value to use in figure 6?

Line 249: How does lower average GRs in Puijo affect the analysis? Lower GR gives larger time difference between J7 and J3, mean that inaccuracies in coagulation sinks and neglecting of time dependence of some quantities plays a larger role. Discuss.

Technical Corrections:

Line 35: commas needed around 'at several locations' Line 134: 'used a parabolic differentiation method ON the measured number concentration' instead of TO Line 276: new paragraph needed for "the ultimate aim of this work"

---

## Author Comment (AC1) · 12 Jul 2017

We thank referee #1 for his/her thorough review of our manuscript. The comments were extremely valuable and we have redone most of the analysis based on them. The main weakness of the previous submitted version was the poor performance in estimating the growth rate with the mode fitting method, which further meant poorly predicted time lag and poor performance in the time resolved formation rate comparison. We have now reanalyzed the growth rates with the so-called maximum concentration method and the results are overall muck better. All figures and table 1 are modified accordingly,

and we also removed the old Figure 3 comparing time-lags, as we believe that it is unnecessary in the new version. We also removed the standard error color coding, related to the uncertainties when determining GR, from Figure 1.

Below we give our detailed responses to the referee's comments.

General Comments:
*A thorough, data-based evaluation of whether particle formation rates can be extrapolated from measurements at larger sizes, as attempted by this paper, is vital for the aerosol community as so many data exist with only larger size information available. While the method used to tackle this problem is valid and useful, the evaluation requires development and more nuanced analysis before the substantial conclusions stated in the paper can fairly be reached (see below for specific comments).*

Specific Comments:
*A major assumption, that the two measurement sites are directly comparable with the method used for extrapolating nucleation rates, is made in the paper. Kurten et al showed that the method used, while valid in many circumstances, may not be valid for situations where pre-existing populations of aerosols do not dominate the coagulation sink and newly formed particles play a larger role in this sink. The differences in background aerosols at the two sites should be discussed in relation to this. Differences between the two site may also influence the magnitude of growth rates and coagulation sinks, which may affect the accuracy of the J extrapolation, which should be addressed.*

The referee has a valid point that, generally, differences in aerosol dynamics (i.e which processes are dominant for the growth and loss of the newly formed particles) between two sites could potentially lead to erroneous conclusions when comparing the scaled J values. However, here this is not the case. The background distributions in our two sites are quite similar, both in total concentration and mode location. The mean values of CoagS of 7 nm particles are $5.3077 \cdot 10^{-5}$ s$^{-1}$ and $5.3272 \cdot 10^{-5}$ s$^{-1}$ in Hyytiälä (mean value of all analyzed NPF event days during 2002-2012) and Puijo (mean value of all

analyzed NPF event days during 2007-2015), respectively. For both sites the nucleation mode concentrations are so small that both the contribution of self-coagulation on growth as well as the contribution of newly formed particles on the sink are negligible. The contribution of particles of different sizes on the sink has been investigated by Lehtinen et al. (Boreal Environment Research 8, p. 405-411, 2003 – see fig. 3) for Hyytiälä size distributions. Particles below ca. 50 nm in diameter have typically negligible effect on condensation/coagulation sinks. As Puijo size distributions are very similar, this conclusion holds also there.

We added after Eq. 1 on line 110: "Lehtinen et al. (2003) studied the contribution of particles of different sizes to the condensation sink at Hyytiälä and found that particles below 50 nm in diameter have typically negligible contribution. This is a reasonable assumption at Puijo also as the concentrations and size distributions are similar to those at Hyytiälä. The mean values of CoagS of 7 nm particles are 5.31e-5 s-1 and 5.33e-5 s-1 in Hyytiälä (event days during 2002-2012) and Puijo (event days during 2007-2015), respectively."

*Line 59: The assumptions that the coagulation sink is time independent and the growth rate size independent should be more fully investigates. Kurten et al. highlights the possibility and affect of time dependent coagulation sinks. If this is not a problem for these two sites it should be explained more explicitly.*

This is true and in our analysis we do not take time dependence of CoagS and GR into account. This is, however, intentional from our part since we wish to follow the procedure of Kulmala et al. (Nature Protocols) in order to analyze formation rates consistently with most other studies previously analyzed.

Below in Figure 1 we show the median diurnal variation for CoagS (3nm). We also added in the revised manuscript Fig. 2 the CoagS time evolution for each of the three example events. It is clear that there may be significant time evolution in the CoagS/GR term of Equation 1, which is of course one of the key reasons why the simple approximation equation is not perfect.

The mentioned assumptions are mentioned in the text after Eq. 1, but to clarify this, we

added to the conclusions (lines 253-256 of the revised manuscript): "when considering detailed daily time evolution, the agreement is not as good. This is caused by three main things: 1. there are significant fluctuations in experimental size distribution data, 2. the extrapolation method assumes a constant value for CoagS/GR, and 3. there is a time lag between J3 and J7 and a poor estimation of the growth rate GR results in comparing values at different times."

*Line 75: some discussion of how the different environments of Hyytiala and Kuopio affect the average size distributions and patterns of nucleation would be helpful here. This can affect how accurately equation 1 can be applied. Equation 1 assumes that the coagulation sink is dominated by larger pre-existing populations, which is less applicable in cleaner environments. If the Hyytiala environment is much cleaner, for example, than Kuopio, then the two situations are less comparable for this method of calculating formation rates. This assumption is mentioned on line 114, but it's validity for both situations requires further discussion.*

See our reply to the first comment above: nucleation mode has negligible contribution to CoagS both in Hyytiälä and in Puijo. The coagulation sink levels in Hyytiälä and Puijo are very similar.

*Line 145: averaging of m and CoagS(d1) between t and t' may be inaccurate, especially for high Js and low GRs – is the any indication of this in the data? How much do m and CoagS(d1) differ between t and t'?*

The median variation of the CoagS over all the analyzed NPF event days is shown in Figure 1 below (not included in the revised manuscript). To illustrate the temporal variation we added into Fig. 2 of the revised manuscript also the time evolution of CoagS for the selected three NPF events. This variation naturally limits the validity of the constant CoagS assumption when applying Eq. 1.

We also added at the discussion of Fig. 1 (lines 196-199 of the revised manuscript): "This is caused by three main things: 1. there are significant fluctuations in experimental size distribution data, 2. the extrapolation method assumes a constant value for

CoagS/GR, and 3 there is a time lag between J3 and J7 and a poor estimation of the growth rate GR results in comparing values at different times."

The effect of the CoagS variation on $m$, and further on $\gamma$ is, however, minor. We now mention this when discussing the result in Figure 1 (lines 199-200 of the revised manuscript): "The variation of CoagS with time also affects $m$ and $\gamma$ in equation 1. This is, however, negligible as CoagS(7 nm)/CoagS(3 nm) is a very weak function of time."

*Line 172: This discussion of how well J3,est and J3,obs agree needs further development. Suggest removing qualitative judgement of 'reasonably well', and leaving only quantitative measurements of this. While the 0.78 correlation coefficient is helpful, the (linear?) fit result that this relates to would give a better measurement of the systematic difference between the two, this needs to be given here and on figure 1. This would then also quantify the following assertion that equation 4 overestimates the formation rate.*

We have now removed 'reasonably well' and added the linear regression line to Fig. 1 of the revised manuscript, as the referee suggested. We also now show the results for both growth rate ranges studied: 3-10 nm and 7-20 nm. Note that, as mentioned at the beginning, we have redone all calculations – now using the maximum concentration method to determine the growth rate. Now, especially the time resolved comparison shows a much better result than previously. The slopes and correlation coefficients for the regression lines are 0.90 and 0.90 for the mean J3 values and 0.87 and 0.83 for the time resolved ones, respectively. There is a slight overestimation bias for small and underestimation for large J3 values. We have added this to the discussion of Fig. 1 in the revised manuscript (beginning of Section 3.1): "Figure 1 shows the comparison of estimated formation rates J3,est (Eq. (4)) with the observed ones J3,obs, as calculated directly from the measured size distribution evolution according to Eq. (2) in Hyytiälä. In the top figures, the size range 3-10 nm is used to evaluate the growth rate, in the bottom ones 7-20 nm. We analyzed 65 NPF event days for which the formation and growth rates could be quantified. Each data point in Figures 1-b and 1-d represents the arithmetic mean of the 3-nm particle formation rates (J3,est and J3,obs) for a single

NPF day during the time window from 07:00 to 19:00 local time. The mean is also a measure of the total particle production strength of each event. The results show that, when using GR in the range 3-10 nm, the estimated mean J3,est values correlate with J3,obs with a correlation coefficient of 0.90 and a slope of 0.90 using bilinear fitting. Furthermore, 91% of estimated J3,est are within a factor of two of the observed J3,obs. The corresponding numbers when using GR in the range 7-20 nm are 0.92 (correlation coefficient), 0.87 (slope) and 93% (J3,est within factor of 2 from J3,obs). Equation (4) seems to have a tendency of slightly overestimating the formation rate of 3-nm particles. There is not much difference in the results with different GR size ranges. The total means of J3,obs and J3,est (not shown in the figure) calculated using GR3-10 are 0.57 and 0.61 cm-3 s-1, respectively, confirming the tendency of Eq. (4) in slightly overestimating the 3-nm particle formation rates."

*Line 175: Standard deviation should be given along with the daily means. These means are taken over a long period of time, during which I suspect J varies quite a bit. If J does vary a lot of this time period, then taking a daily mean is not very meaningful.*

We chose not to add the standard deviations to the plot with means as the time resolved all-data-plot reveals the variation in J-values. The daily mean is meaningful in the sense that it is a measure of the overall strength of a nucleation event. Another variable choice would be the total number of particles produced at some size, but as most of the existing literature reports rates, we chose this approach.

*Line 184: The quoted daily median values of J are very small. Where in the day do they actually occur? Is it actually before a significant nucleation event occurs? If so these values are not very meaningful – suggest either cutting data to only encompass the nucleation event or finding a more meaningful statistic here.*

The median values were calculated over the same time window (07-19) as the mean values given in the text. We, however, agree with the referee that median is not necessarily the best statistic to use here, and decided to remove the median values from the text altogether.

*Line 188: I would argue that reduction of percentage of points within a factor 2 of Jobs from 85% to 78% still reasonably significant and could indeed indicate that GR3-10 different and more accurate than GR7-20, which has strong implications for the conclusion that it's ok to use this GR in extrapolating results from Puijo. It would be more meaningful here to look at the fit equation again rather than simply correlation and percentage within factor 2 to understand this difference better.*

We have now fitted the scatter plots of J3,est vs. J3,obs as suggested by the referee, and explained before, and show the fits in Figure 1 of the revised manuscript. Using the maximum-concentration GR for calculating J3,est, the effect of using GR7-20 instead of GR3-10 is much smaller and overall the results are better (91% and 93% of the daily mean J3,est are within factor of 2 from J3,obs).

*Lines 191-195: Would prefer to see a full comparison of difference between observed and calculated Js here using GR3-10, GR3-7 and GR7-20, as well as a developed discussion of the degree of agreement and implications of this for using this method for J extrapolation. "Did not affect the results . . .by much" is too qualitative and glosses over what could be an important result here.*

This is an excellent suggestion, and based on it we expanded Figure 1 in the revised manuscript to include results using both size ranges for the growth rate GR calculation, 3-10 nm and 7-20 nm. Figure 2 below shows the new Fig. 1 of the revised manuscript, and we also made the discussion related to it more quantitative.

*Lines 196-203: The lack of correlation on temporarily resolved data here may indicate that the growth rates are wrong – this should be discussed here. It could also be because, by taking averages over long events where J varies significantly, the correlation seen early was simply an artifact of such heavy 'smoothing'. Is there another, meaningful measure of J (e.g. peak J of an event) that could be compared to asses this?*

The reanalysis of growth rates with the maximum concentration method has improved the time resolved results significantly as explained before (see figure above). We also choose to stick with looking at mean formation rates as they are a good measure of

overall event strength.

*Lines 205-208: Given the lack of correlation for time resolved Js, testing the affect of different GRs here does not have much meaning – suggest leaving this out completely.*

Regarding the original version of the manuscript, we agree with the referee. However, now with much improved performance with respect to time resolved formation rates, this comparison is meaningful, we think.

*Lines 211-212: "For some NPF days, the estimated time dependence and values of Jest are in fairly good agreement with those of observed Jobs." This statement needs better quantification to be of value. What proportion of days (since we're looking at a relatively small number of event, suggest quoting both total number of events examined and number of those with time dependence and value agreement here instead of just a percentage). How 'fairly good agreement' was judged needs explanation Also, are there distinguishing features of this sub-group where agreement is good? E.g. slow growth, classic 'banana' nucleation pattern?*

With the new analysis for GR these figures (see Fig. 2 of the revised manuscript) have now also changed – and the results are generally much better. Still, the motivation behind Fig. 2 is the same: we show why for some events (and estimated GR) the analysis works better and for some worse. Thus we also chose not to give quantified information on the comparisons of Fig. 2. We explain this now clearly with the discussion of Fig. 2.

*Line 213: quantify 'most of those days'*

As the performance related to time resolved data is now much better we have modified this part of the text: "However, the time-dependency of J3,est is not consistent with J3,obs for some most of the days and, instead, typically the J3,est peak occurs earlier than the J3,obs peak (see e.g. Figure 2-e), indicating that our method of estimating GR is not always satisfactory perfect and typically underestimates the GR values."

*Line 221: why does this burst of particles of 3-7nm occur and not then grow? Is this*

*indicated by the calculated GR and coagulation sink? Or is it perhaps a transport arti-fact? If it is the later it should be removed from the analysis as it is not nucleation. If it's the former then the equation used to calculate J3 should be able to handle it. Therefore this needs full investigation and explanation.*

We investigated the event in more detail and found that it is a transport artifact. This is, of course, one of the general problems when analyzing events measured at one fixed location. In the figures we do not see the same aerosol growing, but particles formed at various location appearing at the measurement site at various stages of their growth. If we have a large enough homogeneous region of similar formation and growth, there is no problem. However, if there are inhomogeneities and the air mass transport direction changes during an event, we see dynamics as in fig. 2c and f. As this day was still classified as an event according to the protocol by Kulmala et al., we chose to include it – also to show what kind of challenges there can be.

We added at the end of section 3.1: "This is one of the general problems when an-alyzing events measured at one fixed location. We do not observe the same aerosol growing, but particles formed at various location appear at the measurement site at various stages of their growth. If we have a large enough homogeneous region of sim-ilar formation and growth, there is no problem. However, if there are inhomogenities and the air mass transport direction changes during an event, we see dynamics as in fig. 2c and f."

*Lines 225-226: Estimated time-lag longer than observed time-lag indicates that the GR used is too low, which has implications for the calculated J and the time-dependence of the nucleation event. Can this explain the poor ability of this method to reproduce the time-evolution of nucleation events? This should be discussed.*

Yes, true. As mentioned before, now the growth rate analysis has been redone using the maximum-concentration method, and this approach gives much better results.

*Line 227: 15 days out of how many in total?*

We removed the old Fig. 3 from the revised version of the manuscript, as we think it is

now not necessary and the text related to that figure was also removed.

*Line 227: 1.5 hours difference: what percentage of the total time lag is this?*

With our new results, having improved growth rates as well as a much better match between the observed and estimated formation rates, we decided to remove (the old) Fig. 3 as well as related text.

*Line 229: quantify 'reasonably good accuracy'*

For the J3,est calculated using GR3-10 from maximum-concentration method, the frac-tion of data points J3,est vs. J3,obs which are within factor of 2 is now 91% (67 out of 74 events) for the daily mean values. For all the 10-min data points it is 58% within factor of 2 (77% within factor of 3, and 84% within factor of 4). We now focus on the numbers (when discussing our results) and leave out these more vague statements.

*Line 244: This monotonic increase in number of event days per year with time is indeed worth noting. Is this because of improvements in instrumentation/data quality? Change of activity or climate in the local area? Some discussion warranted. Do other things, such as total number of nucleation mode particles, size of coagulation sink, or anything else also monotonically change over this time period that might indicate why this is happening?*

We took a look at this once again and now feel that this trend is far too short to be considered a significant trend. In Hyytiälä this time period shows a decrease both in SO2 and CS, which have opposing effects on nucleation event probability. As we cannot quantify/justify such a trend with our supporting measurements we decided to remove the sentence.

*Line 245: Given the lack of correlation shown early between median J3 est and obs, using J3 est here for analysis does not seem justified. Surely mean J, where some correlation between estimated and observed values was calculated is the value to use in figure 6?*

This was a typo. The presented values are means, which makes much more sense.

*Line 249: How does lower average GRs in Puijo affect the analysis? Lower GR gives larger time difference between J7 and J3, mean that inaccuracies in coagulation sinks and neglecting of time dependence of some quantities plays a larger role. Discuss.*

The referee has a good point here. As we reanalyzed the growth rates using the maximum-concentration method, the average GR in Puijo is actually slightly higher than in Hyytiälä. We added to the revised manuscript (lines 236-237): "As the growth rates in Puijo are on average higher than in Hyytiälä, there is less time needed for the particles to grow from 3 to 7 nm. This means that our assumption of time independent growth rate and coagulation sink during growth should hold in Puijo as good as in Hyytiälä."

Technical Corrections:

*Line 35: commas needed around 'at several locations'*
Corrected.

*Line 134: 'used a parabolic differentiation method ON the measured number concentration' instead of TO*
Corrected.

*Line 276: new paragraph needed for "the ultimate aim of this work"*
Corrected.

[Figure]

**Fig. 1.** The median diurnal variation of coagulation sink of 3nm particles for all the NPF events analyzed in this study in Hyytiälä. The error bars indicate the 25th and 75th percentiles of the CoagS data.

[Figure]

**Fig. 2.** Estimated J3,est and observed J3,obs formation rates of 3 nm particles in Hyytiälä, calculated using GR by maximum-concentration method. This figure is included in the revised manuscript as Figure 1.

---

## Author Comment (AC2) · 12 Jul 2017

We thank Referee #2 for his/her thorough review of our manuscript. The comments were extremely valuable and we have redone most of the analysis based on them. The main weakness of the previous submitted version was the poor performance in estimating the growth rate with the mode fitting method, which further meant poorly predicted time lag and poor performance in the time resolved formation rate comparison. We have now reanalyzed the growth rates with the so-called maximum concentration method and the results are overall muck better. All figures and table 1 are modified

accordingly, and we also removed the old Figure 3 comparing time-lags, as we believe that it is unnecessary in the new version. We also removed the standard error color coding, related to the uncertainties when determining GR, from Figure 1.

Below we give our detailed responses to the referee's comments.

GENERAL
*The manuscript is basically suitable for ACP but a few important points need to be addressed first. I'll rate this "major revision" for now but remain skeptical that all concerns can be addressed to my satisfaction.*

*1. The authors present a method to estimate formation rates for smaller particles based on measurements of the formation rate for larger particles. This is no doubt necessary to compare measurements with different instruments and to gain information on the actual nucleation rate which happens at sizes which are often outside the measurement range. However, a few years back, Kulmala et al. have published a sort of how-to guide on nucleation measurements in Nature Protocols. And that looks very, VERY similar to what is presented in this manuscript while the text reads as if something new is shown. So my question would be: What is actually new in the approach that the authors present? Or is this just another application of the same formula that has been used in lots and lots of papers for quite a number of years? If this were the case, the manuscript's content would be very slim indeed (and all the description of methods used obsolete) and one would have to ask if just running an old formula on a new set of data would justify a scientific publication.*
The method used in our manuscript for scaling the formation rates is indeed the same as described by Kulmala et al. in their Nature Protocols paper. While the method itself is not new, it has not been tested with atmospheric particle number size distribution data before (to our knowledge), although widely used in e.g. global modelling of aerosol dynamics. In our work, we present comparisons with the scaled formation rates to those calculated directly from the measurement data, and thus evaluate the applicability of this method with real atmospheric data.

[Figure]

*2. The method to determine GR obviously doesn't work as the authors themselves point out. That is not a surprise since mode-fitting at the edge of a size distribution is always a bad idea. However, there are other methods. How can you justify using a method that clearly produces bad results while there are alternatives available? Sure, other methods typically are much more labour-intensive but given the current state of affairs it seems quite clear to me that other approaches MUST be employed. At least a preliminary test on a smaller subset of the data is absolutely and totally necessary.*

As the referee notes, the main weakness of the previous submitted version was the poor performance in estimating the growth rate with the mode fitting method, which further meant poorly predicted time lag and poor performance in the time resolved formation rate comparison. We have now reanalyzed the growth rates with the so-called maximum concentration method and the results are overall much better (the updated Fig. 1 included in the revised manuscript is shown as Figure 2 in our replies to Referee #1). All the figures and Table 1 are modified accordingly, and we also removed the old Fig. 3 comparing time-lags, as we believe that it is unnecessary in the new version.

*3. I do wonder how there can be only 65 days with good enough data from 12 or 13 years of Hyytiälä observations. Assuming 12 full years à 365 days and an NPF frequency of 23% (Nieminen et al., 2014), that's a tiny 6.5% of all nucleation events observed during that time (about 1000). How can that be? Are we supposed to believe that one of the longest and probably the most published data set of aerosol size distributions is actually total crap? I mean, I have worked with DMPS and SMPS data quite a bit but never ever has the data been so terrible that a proper analysis was possible for less than 10% of events. And even if I was to accept this low percentage (which I can't and won't) then the question arises if this kind of cherry-picking doesn't introduce a bias into the analysis that would make all and any results highly questionable.*

The number of days included in our analysis is not limited by data availability, but rather the criteria of the NPF event analysis: the growing nucleation mode needs to be clearly observable for several hours (i.e. no changes in air masses). Typically in Hyytiälä the

number of these "well-behaved" NPF events is around 10% of all days (Dal Maso et al., 2005). In the Dal Maso classification the NPF events in Hyytiälä are classified as Ia, Ib, and II. For the analysis of this manuscript, we only chose class Ia NPF events, producing the number of events analyzed here. This strict selection of NPF events was done because we wanted to eliminate the possible effect of e.g. changes in air masses in our results.

CONTENTS
*line 31f "(e.g. Almeida et al., 2013; Berndt et al., 2014; Kirkby et al., 2016)" –> Bianchi et al., 2016 should probably be added there.*
We added this reference.

*line 51 "we aim to estimate 3 nm particle formation rates" –> why? the 3 nm limit has no physical meaning, it's just a tradition born out of instrumental limitations from two decades back. i understand that you do that for the hyytiälä data since the point is testing the approach. but for puijo?*
The referee is correct that nowadays the aerosol instruments are able to measure particles down to 1.5 nm. However, since the vast majority of particle formation rates reported in the literature is at 3 nm, we chose to scale also the Puijo data to this size. That way the Puijo results can be more directly compared to observations at other sites.

*line 88ff –> this whole section is useless if this is the same approach as in the Nature Protocol*
The method is the same as described in the Nature Protocol paper, however we feel that presenting the method in our manuscript makes it easier for the user to read our paper. One other reason for this choice is the need to use growth rates in the equations. For Hyytiälä data, growth rates are available down to 3 nm while for Puijo only above 7 nm. Thus we wanted to make clear in the equations what size ranges we are using.

*line 164 "the size dependence of the growth rate in the range 3-20 nm is typically weak"*

*–> really? or is this just an artefact of the GR approach not working (which we know is true). certainly you could cite some previous studies that have found this; hyytiälä isn't exactly under-studied after all.*

The referee has a valid point here. As we recalculated the growth rates using maximum-concentration method, there is indeed a size-dependency in the growth rates, as is shown in Figure 1 below (the red line shows the median, the edges of the box the 25th and 75th percentiles, and the error bars the 10th and 90th percentiles of the GR values of NPF events in Hyytiälä; the red data points are all GR values which are larger than the 90th percentile). However, this does not affect greatly the correlation between J3,obs and J3,est (calculated either using GR3-10 or GR7-20), as can be seen from Fig. 1 in the revised manuscript.

*line 173 "85 %" –> wasn't it 84 in the abstract?*

All the results related to comparison of the estimated and observed formation rates are updated in the revised manuscript according to the new J3,est values, which are calculated using the GR from maximum-concentration method. The correlation coefficient between J3,est and J3,obs is now 0.90 and 91% of the J3,est daily mean values are within factor of 2 from J3,obs.

*line 181ff –> the whole median thing seems a bit silly. i mean, you take the median over 12 hours during most of which there is no formation of 3 nm particles. Of course the result will be close to 0 (as it is). a pointless exercise which tells us nothing.*

We agree, and therefore we have left the median values out from the revised manuscript.

*line 201ff, line 210ff, and lots of other places –> i won't comment on the GR stuff here, see general comments above.*

As mentioned previously, we chose a better method for the GR analysis, and the results improved a lot.

*line 216f "the days during which a clear peak in each of the [different] time evolution*

*curves could be observed (39 days out of 65 days)" –> 39 out of 65 sounds quite good. but really it is 39 out of 1000, and that is not acceptable.*

As mentioned before we were very selective with the events analyzes and chose only the so-called 1A events based on the Dal Maso et al. classification. This was done in order to be sure that e.g. small changes in airmasses would not influence the results of the J estimation.

*line 224 "It can be also concluded that visual inspection of the data is still valuable" –> that's sound advise that you might want to follow with regard to the GR business.*

So true. As we redid our analysis with the improved growth rate analysis, we also checked each analyzed event visually.

*line 227 "There are 15 NPF days for which the estimated time-lag is within 1.5 hours of the observed time-lag." –> please take a moment and think what you have written there. with an average GR of roughly 4 nm/h, the average time-lag should be around 1 hour, right? that 15 cases lie within 1.5 hours is no proof that the method works sometimes but rather that the method does not work AT ALL.*

With our new results, having improved growth rates as well as a much better match between the observed and estimated formation rates, we decided to remove (the old) figure 3 as well as related text.

*line 227ff "Overall these results from analyzing Hyytiälä data show that Eq. (4) can be used to estimate the mean formation rates of 3-nm particles with reasonably good accuracy." –> but maybe things could be much better with an improved determination of GR?*

Yes, true. Now they are.
* * *
[Figure]

[Figure]

[Figure]

**Fig. 1.** Statistics of the growth rates in 3-10 nm and 7-20 nm size ranges in Hyytiälä, calculated by mode-fitting (left panel) and maximum-concentration (right panel) methods.

---

## Referee Report (RR1)

Baranizadeh et al review

**General Comments**
The comments on the initial review have mostly been well addressed (see specific comments for details) and the improved growth rate calculations have greatly enhanced the value of this analysis. The presentation of results and analysis is much improved. The analysis now allows substantive conclusions to be made, which the paper does not yet do. In addition, there is a puzzling result of the growth rates at larger sizes providing better calculated nucleation rate correlation with measurements than those at the smaller, relevant sizes. This should be looked into before the analysis is considered complete.

**Comments**

Response to comment on line 221 – how was this identified as a transport artifact?

Line 32 – dispute that 'actual mechanisms' of nucleation remain unknown, experiments such as CLOUD have shown then 'actual mechanisms' of many types of nucleation in great detail.

Line 40 – even with sub-2nm cut-offs Js still have to be approximated – yes but explain, not all readers will be familiar with this

Fig 1 – agreement is consistently better with GR7-10nm, this seems odd. How do the GR3-10 and GR7-20 compare – there must be a systematic different, or an extra uncertainty in the GR3-10 to explain this, or something wrong the J equation that means the more removed 7-20nm GR is compensating for an error – needs to be evaluated

Lines 237-240 - quantify 'most' and 'some' and 'fairly-good agreement' with statistics

Line 283 'should hold as good as in Hyytiala' – not really accurate, if GRs are faster in Puijo, then the GR uncertainties relating to time dependence on change in coag sink will affect J less. Technical note 'as well as in Hyytiala' instead of 'as good as in Hyytiala'

Line 300-303: this sentence is not very clear on the actual cause of the poor time evolution agreement. The effect of the 3 listed factors on the time evolution needs to be explained better.

Lines 309-310: Statement about Asmi's reported J7s at Pallas requires a clearer link to the work in this paper if it is to be included.

Lines 311-313: Agree about the challenges faced in calculating J3 from J7, however the paper would be of much more use if a quantitative statement about the utility of the presented method and analysis were made. This study can and should be used to make a quantitative evaluation of the utility of this method to calculate J3. Either it is or is not worthwhile, and a number can be put on the accuracy of the method based on the data presented here. The author may wish to consider putting this in the broader context of things e.g. when put into climate models, what is

the general sensitivity in CCN number concentration or even CN3 or 10 to a factor 2 change in J?

**Purely Technical Comments**
Line 21 were -> was

---

## Author Response (AR2)

We thank both referees for their comments on our revised manuscript. Below we give our response to each of the comments, and indicate if anything was changed in the manuscript based on the comments. The marked-up revised manuscript is included after our replies to the referee comments.

Anonymous Referee #1

*The comments on the initial review have mostly been well addressed (see specific comments for details) and the improved growth rate calculations have greatly enhanced the value of this analysis. The presentation of results and analysis is much improved. The analysis now allows substantive conclusions to be made, which the paper does not yet do. In addition, there is a puzzling result of the growth rates at larger sizes providing better calculated nucleation rate correlation with measurements than those at the smaller, relevant sizes. This should be looked into before the analysis is considered complete.*

Comments
*Response to comment on line 221 – how was this identified as a transport artifact?*
This is based on visual inspection of the measured aerosol size distributions, where we see sudden changes both in the nucleation and accumulation mode around 12 o'clock. Also the local wind direction changes during the same time as these change in the aerosol size distributions is observed, suggesting that different air masses are sampled.

*Line 32 – dispute that 'actual mechanisms' of nucleation remain unknown, experiments such as CLOUD have shown then 'actual mechanisms' of many types of nucleation in great detail.*
This sentence refers to nucleation in the atmosphere, where it might be still argued that the exact nucleation mechanisms in different environments are not yet fully understood. We modified the text in the manuscript: "However, several features of atmospheric nucleation including the actual mechanism in different environments …"

*Line 40 – even with sub-2nm cut-offs Js still have to be approximated – yes but explain, not all readers will be familiar with this*
We modified this sentence to: "… the determination of nucleation rates still involves approximation, e.g. due to composition dependent detection efficiencies and high loss rates of the smallest particles."

*Fig 1 – agreement is consistently better with GR7-10nm, this seems odd. How do the GR3-10 and GR7-20 compare – there must be a systematic different, or an extra uncertainty in the GR3-10 to explain this, or something wrong the J equation that means the more removed 7-20nm GR is compensating for an error – needs to be evaluated*
It is true that the $J_{3,est}$ calculated using GR7-20 seems to have slightly better agreement with $J_{3,obs}$ than those calculated using GR3-10. However, this should not be over-interpreted: we evaluated the statistical significance of all the correlations of $J_{3,est}$ vs. $J_{3,obs}$ presented in Fig. 1. We performed bootstrap analysis by re-sampling the $J_{3,est}$ vs. $J_{3,obs}$ data sets with substitution, and calculating the correlation coefficient and the fitting parameters *a* and *b* for this re-sampled data set. This bootstrap process was repeated 1000 times to obtain the distribution of the correlation coefficients and fitting parameters, and the confidence intervals were then determined by the 5th and 95th percentiles of each distribution. We now give the confidence intervals for the correlation coefficient and fitting parameters in the modified Fig. 1. As the confidence intervals of all the correlation coefficients and fitting parameters are overlapping, we can conclude that the $J_{3,est}$ calculated using GR7-20 is not statistically significantly better from $J_{3,est}$ calculated using GR3-10, rather than both give similar agreement with $J_{3,obs}$.

*Lines 237-240 - quantify 'most' and 'some' and 'fairly-good agreement' with statistics*
We added the relevant statistics into the revised manuscript: "For most of the NPF days (81% of the days) the estimated time-dependence of $J_{3,est}$ (or time-lag between 3-nm and 7-nm particle formation rates) is within one hour of the observed $J_{3,obs}$, and the values of $J_{3,est}$ are in fairly-good agreement with $J_{3,obs}$ (see e.g. Figure 2-d). However, the time-dependency of $J_{3,est}$ is not consistent with $J_{3,obs}$ for some of the days (19% of the days have larger than one hour time difference between $J_{3,est}$ and $J_{3,obs}$) and, instead, typically the $J_{3,est}$ peak occurs earlier than the $J_{3,obs}$ peak (see e.g. Figure 2-e)."

*Line 283 'should hold as good as in Hyytiala' – not really accurate, if GRs are faster in Puijo, then the GR uncertainties relating to time dependence on change in coag sink will affect J less. Technical note 'as well as in Hyytiala' instead of 'as good as in Hyytiala'*
We changed this to "as well as in Hyytiälä" according to the referees suggestion.

*Line 300-303: this sentence is not very clear on the actual cause of the poor time evolution agreement. The effect of the 3 listed factors on the time evolution needs to be explained better.*
We modified the sentence to: "This is caused by three main things. First, there are significant fluctuations in experimental size distribution data due to e.g. changes in the sampled airmasses. This kind of fluctuations are not taken into account in Eq. 1. Second, the extrapolation method assumes a constant value for CoagS/GR. If this is not the case, it affects both the time evolution (determined by GR) and the magnitude of the estimated J (determined by the ratio CoagS/GR). Third, there is a time lag between $J_3$ and $J_7$, and a poor estimation of the growth rate *GR* results in comparing values at different times."

*Lines 309-310: Statement about Asmi's reported J7s at Pallas requires a clearer link to the work in this paper if it is to be included.*
We decided to remove the reference to Asmi et al., and compare the Puijo results only to those from Hyytiälä.

*Lines 311-313: Agree about the challenges faced in calculating J3 from J7, however the paper would be of much more use if a quantitative statement about the utility of the presented method and analysis were made. This study can and should be used to make a quantitative evaluation of the utility of this method to calculate J3. Either it is or is not worthwhile, and a number can be put on the accuracy of the method based on the data presented here. The author may wish to consider putting this in the broader context of things e.g. when put into climate models, what is the general sensitivity in CCN number concentration or even CN3 or 10 to a factor 2 change in J?*
The comparison of the $J_{3,est}$ and $J_{3,obs}$ from Hyytiälä presented in Fig 1. shows that the daily mean values can be estimated within factor of 2 in over 90% of the cases. This is also written in both the Conclusions (lines 247-248 of the revised manuscript) and the Abstract (lines 17-18). We feel that these statements clearly demonstrate and quantify the usefulness of the method.
The referee has a good suggestion to evaluate the implications that our results would have on global modelling studies. However, we feel that it is out of scope of the current manuscript.

Purely Technical Comments
*Line 21 were -> was*
Corrected.

Anonymous Referee #3

*New particle formation has been demonstrated to play important roles in air quality and climate change. Measurement of particle size distribution is the basis of most relevant studies. 3 nm was recognized as one critical size in terms of new particle formation. But a considerable amount of measurements on particle size distributions do not extent to 3 nm but instead start at 6 or 7 or 10 nm. Therefore, developing a convincing method to estimate the 3 nm particle formation rates from larger size is crucial to build a worldwide, comparable data set, which will be helpful to understand the mechanism of new particle formation and evaluate their roles in climate. In this study, the authors extrapolated the formation rates at 7 nm (J7) down to 3 nm (J3) at SMEAR IV station based on an approximate solution to the aerosol general dynamic equation. Data from SMEAR II station, which extend down to 3 nm, was used to evaluate the method. The manuscript is overall well written and documented. The topic fits well in the scope of ACP. I recommend this manuscript can be published after some revisions.*

One general comment:
*Line 201-217, authors mentioned there are one group of NPF events for which the J3,est and J3,obs are dramatically different. What's the percentage of this kind of events? Are they included in Figure 1? Are this inhomogenities mainly due to the changes of air masses? If yes, an evaluation on the stability of air masses is recommended during one event. It's better to establish a standard method to recognize the NPF events as quantifiable or not but not based on visual observation.*
Figure 1 includes only the well-behaved NPF events, i.e. those events during which there were no obvious changes in air masses or other disturbances which could affect the comparison of $J_{3,est}$ and $J_{3,obs}$.
The method that we followed in classifying the NPF events is based on visual observation of the aerosol size distributions during one day, and as such has been used in many publications on atmospheric NPF and is well documented (Dal Maso et al., 2005; Kulmala et al., 2012). If the referee's suggestion of "establishing a standard method" is about making the method quantifiable or automated, we feel that this task is out of scope of the current work.

Technical comments:
*1. Line 230: add the bracket and stop after "Dal Maso et al., 2005".*
Done.

*2. Please unify the format of units, e.g. 1/s or s-1.*
We checked that all the units are in presented in uniform format throughout the text.

*3. It seems to be not necessary to show the statistical results at Puijo both in Figure 5 and Table 1.*
We decided to present the results both Fig. 5 and Table 1. From Table 1 it is easier for the reader to obtain the relevant numbers, whereas Fig. 5 provides an overview of the seasonal variation of the parameters and their variation within each season.

[revised manuscript text omitted]